# Observed impact of the GNSS clock data rate on Radio Occultation bending angles for Sentinel-6A and COSMIC-2

Sebastiano Padovan[1], Axel von Engeln[1], Saverio Paolella[1], Yago Andres[1], Chad R. Galley[2], Riccardo Notarpietro[1], Veronica Rivas Boscan[1], Francisco Sancho[1], Francisco Martin Alemany[1], Nicolas Morew[1], and Christian Marquardt[1]

[1]European Organisation for the Exploitation of Meteorological Satellites, Darmstadt 64295, Germany
[2]Jet Propulsion Laboratory, California Institute of Technology, Pasadena, CA 91109, USA

**Correspondence:** S. Padovan (sebastiano.padovan@external.eumetsat.int)

**Abstract.**

Radio Occultation (RO) measurements require the tracking of signals from the Global Navigation Satellite System (GNSS) by a Low-Earth-Orbit (LEO) satellite as the signals travel through different layers of the atmosphere. The orbit and clock solutions for the GNSS constellations affect these measurements in two ways: They are needed to obtain a zero-differencing GNSS-based orbit and clock solution for the LEO, and they enter directly the processing of each single RO profile, where the orbit and clock information for the transmitter (GNSS) and receiver (LEO) satellites is required. In this work, we investigate how different GLONASS and GPS orbit and clock solutions affect the statistical properties of RO profiles by comparing our results with forward-modelled bending angle profiles based on data from the European Centre for Medium-Range Weather Forecasts (ECMWF) short-range forecasts. Given that GNSS orbits are relatively smooth, this study focused on the effect of different transmitter clock data rates and we tested the range from 1 to 30 seconds (specifically, the rates of 1, 2, 5, 10 and 30 seconds). The analysis is based on the reprocessing of Sentinel-6A data (four months in 2021, September to December, or about 110k occultations) and of a smaller sample of recent COSMIC-2/FORMOSAT-7 data (August 5th–7th, 2023, or about 9k occultations). We find that at impact heights above about 35 km GLONASS bending angle statistics markedly improve with the use of higher-rate clock information. For GPS, the statistics are better for more recent GPS blocks, and a rate of 5 s provides a marginal improvement over the 30 s rate for all blocks. In the same impact-height range, higher-rate GLONASS clocks also significantly reduce the vertical error correlation. These results are likely the manifestation of the different short-timescale behaviour of the atomic clocks onboard the GPS and GLONASS constellations.

## 1 Introduction

The calculation of bending angle (BA) profiles based on signals from the Global Navigation Satellite System (GNSS) travelling between a GNSS satellite and a Low-Earth-Orbit (LEO) spacecraft through the Earth's atmosphere requires accurate knowledge of the position, velocity and clock of the two space vehicles. In the Radio Occultation (RO) processing described in this work, the GNSS positions and clocks are provided as auxiliary data. They are used explicitly in the computation of each BA (through,

e.g., the Doppler equation, Kursinski et al., 1997) and implicitly, given that they are required to obtain the LEO orbit and receiver clock solutions.

The GNSS space vehicles have revolution periods in excess of 10 hours and their orbits, which are largely controlled by gravitational dynamics, are relatively smooth and are typically provided at a rate of few minutes. The GNSS clocks are needed to synchronize the receiver clock when performing the LEO Precise Orbit Determination (POD). Due to the random stochastic noises that affect GNSS atomic clocks, a smaller sampling interval is required to obtain accurate interpolations, as needed by the bending-angle-retrieval process. As a reference, the Center for Orbit Determination Europe (CODE) currently provides

orbits and clocks with a 5 min and a 30 s sampling rate, respectively, for rapid products (latency of 18 hours, Dach et al., 2023b) and 5 min and both 5 s and 30 s sampling rate for orbits and clocks for final products (latency of two weeks, Dach et al., 2023a). The Jet Propulsion Laboratory (JPL) Global Differential GPS (GDGPS) real-time (RT) GNSS products are available with a 1 min orbit sampling and a 1 s clock sampling. Table 1 provides the relevant information on these and other sets of GNSS products that have been used in this study.

RO represents a high-rate application of GNSS products, since a typical radio occultation event has a duration of few minutes at most. At these time intervals, different GNSS constellations and different clock hardware within a given constellation show a range of clock stabilities (e.g., Hauschild et al., 2013; Griggs et al., 2015). The implication is that a proper description of the behavior of a clock (i.e., to have a model of its evolution), will require a product with a rate that depends on the clock properties. In other words, a more unstable clock hardware benefits from the use of a higher-rate clock correction.

The analysis of Sentinel-6A (S6A) RO data performed at the European Organisation for the Exploitation of Meteorological Satellites (EUMETSAT) using different GNSS data streams provided the motivation for this study, as described in Sect. 2. The experiment set-up is described in Sect. 3. The bulk of the analysis and the results in Sect. 4 are based on a 4-month-long batch of data from the S6A RO instrument (Sect. 4.1) and are complemented by a smaller batch of COSMIC-2/FORMOSAT-7 data (Sect. 4.2). Discussion and conclusions are presented in Sect. 5 and 6, respectively.

## 2   Motivation

The S6A spacecraft is the latest member of the family of altimetry reference missions (e.g., Donlon et al., 2021). For the first time, such a mission is equipped with an RO instrument, which was built by JPL (see also Paolella et al., 2024, this issue). The receiver is connected to a POD antenna, which tracks only GPS signals, and to two occultation antennas tracking both GPS and GLONASS signals, located in the velocity and anti-velocity direction. The entire mission is operated by EUMETSAT. For

the RO instrument, JPL is responsible for the provision of Near-Real Time (NRT) RO data, while EUMETSAT for Non-Time Critical (NTC) RO products. These two operational streams are based on the same level zero data, received by EUMETSAT from the spacecraft through its ground segment and redistributed to JPL. JPL also hosts one of the analysis centers of the international GNSS service (IGS) and, through an operational agreement, provides the auxiliary GNSS data for both RO data streams. The JPL GPS auxiliary products provided to EUMETSAT for the NTC processing come from JPL final stream, with

orbits and clocks at 15m and 30s, respectively. The JPL GLONASS products come from JPL RT stream, but with orbits at 15m

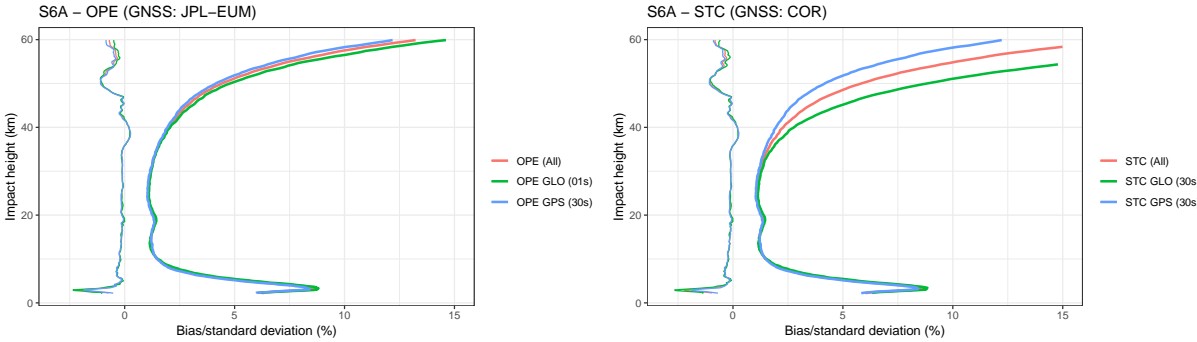

**Figure 1.** Sentinel-6A robust statistics against ECMWF-based short-range forecasts for two data sets for the period Feb 24[th]-Mar 31[st], 2024, corresponding to about 38k single bending angle profiles for each set. **Left**: Operational data stream (OPE), where the POD is based on Bernese 5.2 (Sect. 3.1) and the GNSS auxiliary data are from JPL, with 30 s and 1 s clock rate for GPS and GLONASS, respectively. **Right**: STC offline data stream, where the POD is based on Bernese 5.4 (Sect. 3.1) and the GNSS auxiliary data are CODE rapid products, with clocks at 30 s for both GPS and GLONASS.

and clocks at 1s (we refer to this set, which mixes RT and final products, as JPL-EUM, see Table 1). The NTC data stream (from now on, we refer to this stream as OPE) has a formal timeliness of 60 days, but it is kick-started as soon as the GNSS auxiliary data are available. Typically, EUMETSAT receives this data within two weeks from sensing time.

In addition to the OPE processor for monitoring purposes at EUMETSAT we deployed an offline Short-Time Critical pro-
60 cessor (referred to as STC processor), which uses the CODE rapid products (COR) as GNSS auxiliary data (Dach et al., 2023b), and has a timeliness of about 24 hours. As part of this monitoring activity, for both the STC and the OPE processors, robust statistics are computed for the bias and standard deviation (Maronna et al., 2019) with respect to the forward-modelled bending angle profiles obtained from the temperature, pressure, and humidity profiles extracted from the European Centre for Medium-Range Weather Forecasts (ECMWF) short-range forecasts. The forward modelling is done with the Radio Occultation
Processing Package (ROPP, Culverwell et al., 2015).

The comparison of the statistics between the STC and OPE data streams provides a first indication of the importance of the GNSS clock data rate for the quality of the derived RO products. In Fig. 1 we plot the bias and the standard deviation for both the OPE and the STC processor, using recent S6A data from the period between February 24[th] and March 31[st], 2024, which corresponds to about 38.5k individual bending angle profiles per processor. For this period, the total number of products
obtained with the two processor is slightly different at about the 3% level. The differences can be ascribed to the effect of the different POD SW, the different GNSS auxiliary data used, and the possibility of different amount of level zero data, due to occasional time delays in the satellite's data transmission (this kind of data is received through so-called archive dumps), which could be missed by the STC processor. The bias profiles between the two processors and for the two constellations are self-consistent and hardly distinguishable. For the standard deviation, the situation is different. Both for STC and for OPE the
GPS clock products have a 30 s sampling rate. The different data source (COR for the STC and JPL-EUM for the OPE, see Table 1) does not affect the overall standard deviation for GPS. For GLONASS, the STC processor uses the COR 30 s clocks,

**Table 1.** Summary of the different sets of GNSS products that have been used for this study.

| Abbreviation | GPS rate | | GLONASS rate | | Latency | Description/Notes |
|---|---|---|---|---|---|---|
| | Orbit | Clock | Orbit | Clock | | |
| **GNSS products source** | | | | | | |
| **Center for Orbit Determination Europe (CODE)[1]** | | | | | | |
| COD | 15m/5m | 30s | 15m/5m | 30s | $\sim 14$ days | CODE Finals stream[2] |
| COD 05 s | 15m/5m | 5s | 15m/5m | 5s | $\sim 14$ days | CODE Finals stream with high-rate clocks[2] |
| COR | 15m/5m | 30s | 15m/5m | 30s | $\sim 18$ hours | CODE Rapid stream[3] |
| **Jet Propulsion Laboratory (JPL) Global Differential GPS (GDGPS)[4]** | | | | | | |
| JPL RT | 1m | 1s | 1m | 1s | $\sim 1$ minute | JPL Real-Time stream[5] |
| JPL $dd$ s | 15m | $dd$ s | 15m | $dd$ s | N/A[6] | Downsampled JPL RT (e.g., JPL 10s) |
| JPL-EUM | 15m | 30s | 15m | 1s | $\sim 10$ days[7] | JPL-EUM S6A-RO NTC Operational agreement |

[1] The rate of the CODE orbit products has been increased from 15m to 5m after the adoption of the new IGS20 reference frame (starting on November 27[th], 2022);    [2] For the year YYYY, products are found at http://ftp.aiub.unibe.ch/CODE/YYYY/;

[3] http://ftp.aiub.unibe.ch/CODE/;    [4] https://www.gdgps.net;

[5] https://cddis.nasa.gov/archive/gnss/products/realtime, https://sideshow.jpl.nasa.gov/pub/JPL_GNSS_Products/RealTime;

[6] Since these are a downsampled version of the JPL RT stream, no latency can be defined for them;

[7] These products are not publicly available, but part of an operational agreement between JPL and EUMETSAT.

while the OPE processor is fed with 1 s JPL-EUM clock products. Above about an impact height of 35 km, the GLONASS standard deviation curves (green) are markedly different.

Previous work has pointed out the lesser short-term stability of GLONASS clocks with respect to GPS ones, at least starting with Block IIF satellites (Griggs et al., 2015; Hauschild et al., 2013). The decreased stability does impact the BA uncertainty, as shown in a preliminary analysis of a small batch of COSMIC-2 data by Yao et al. (2023). Less stable clocks would require a higher-rate correction to compensate their noise, and the results of Fig. 1 point into this direction.

Given that the work of Hauschild et al. (2013) and Griggs et al. (2015) is several years old, and both the GPS and GLONASS constellations have been evolving in the meantime, we show in Fig. 2 the modified Allan Deviation (e.g., Griggs et al., 2015) for the 1 s JPL RT clock products for December 31[st], 2021, a date which is part of the main S6A processing campaign below (Sect. 4.1). At that time in the GPS constellation there were 15, 12 and 5 Block II-R, Block II-F and Block III satellites, respectively. The results for the GPS satellites are similar to those of figure 7 of Griggs et al. (2015), with the addition of the group of Block III satellites, which show a better performance on the short timescales ($< 10\,$s), and are similar to the Block II-F between $10^1$ and $10^2$ seconds. The GLONASS constellation was almost completely composed by GLONASS-M satellites, with a single GLONASS-K1, which displays a slightly better performance at the very short timescales.

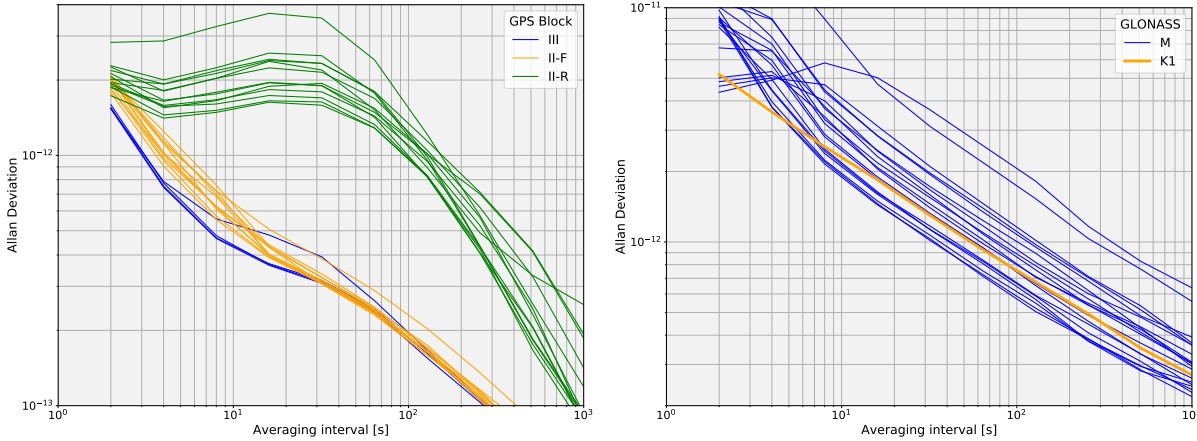

**Figure 2.** Modified Allan Deviation for the GPS (left) and GLONASS (right) constellations for December 31[st], 2021, calculated using JPL RT clock data (Table 1). Please note the different range of the y-axis.

## 3 Set-up of the experiments

Unless otherwise noted, in what follows all the experiments described are based on the re-processing of radio occultation data from the S6A and COSMIC-2/FORMOSAT-7 missions, from level zero up to BA. These two missions feature the same TriG instrument, a hardware component that builds up on the former BlackJack/IGOR receivers and is a POD and RO receiver able to track signals from multiple GNSS constellations (see Esterhuizen et al., 2009, for additional details), equipped with an ultra-stable oscillator (Tien et al., 2012). The level zero data is decoded using a JPL-provided software compliant with the International Traffic in Arms Regulations (ITAR). The RO processing is performed with the EUMETSAT-developed and -maintained Yet Another Radio Occultation Software (YAROS), and detailed information on the processing approach can be found in Paolella et al. (2024, this issue). Here, we provide some details on the POD processing setup and on the employed GNSS auxiliary data.

### 3.1 POD

The LEO orbit and clock solutions are obtained with the latest version of the Bernese GNSS software (version 5.4) using zero-differenced GNSS data (Jäggi et al., 2007; Bock et al., 2011). Bernese 5.4 allows for the modeling of non-gravitational forces on the LEO, namely air-drag and direct and reflected radiation pressure (Mao et al., 2019). To make use of these models, a representation of the shape of the spacecraft and of the material properties of its exterior layers is required. In our simulations we use a macro model for S6A based on Montenbruck et al. (2021), but no shape model is employed for COSMIC-2. Bernese 5.4 has the capability of performing zero-differencing integer ambiguity resolution using the Bias-SINEX products (Villiger et al., 2019; Schaer et al., 2021). However, these products are not part of the operational agreement with JPL and our solutions are based on the float-ambiguity resolution approach. Integer-ambiguity fixing has been shown to perform better than the float-

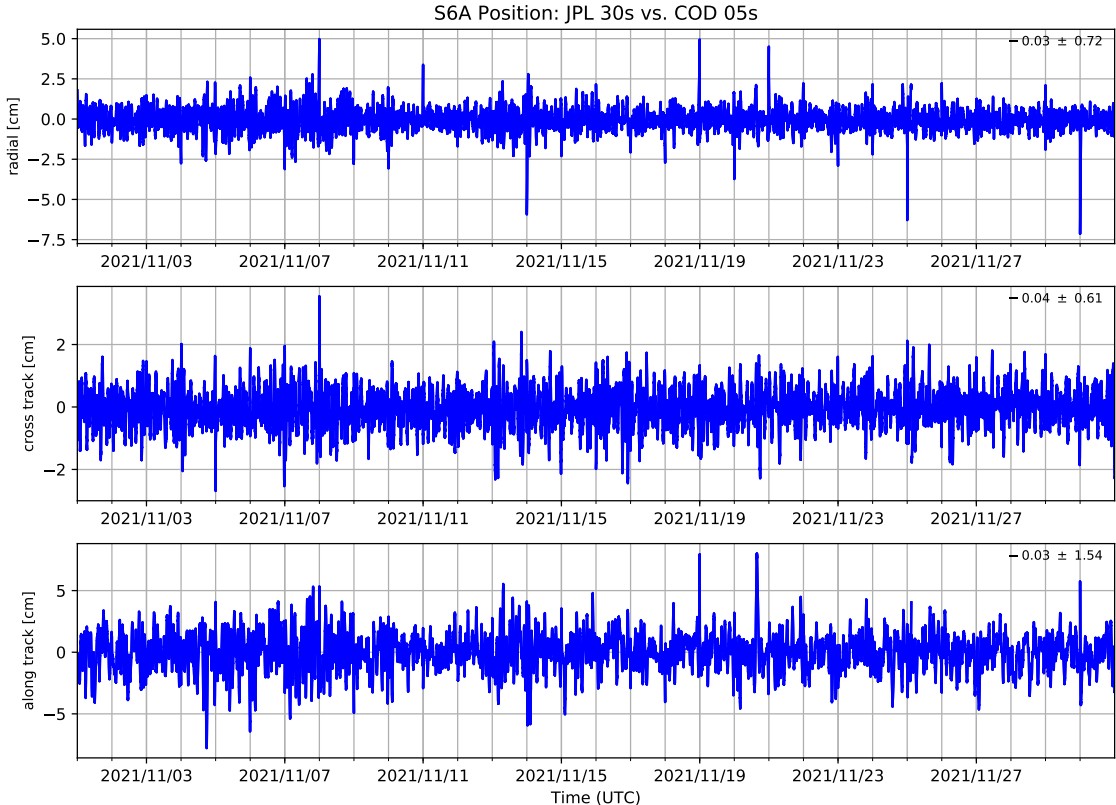

**Figure 3.** Comparison along the radial (top), cross track (middle) and along track (bottom) directions of two S6A orbit solutions for the month of November 2021. In each panel the mean $\pm$ the root mean square (RMS) is indicated. The two solutions are obtained with the same software (Bernese 5.4) but with two different GNSS GPS auxiliary data products (Table 1): JPL 30 s (15 min orbits and 30 s clocks) and CODE 05 s (15 min orbits and 5 s clocks).

ambiguity approach (e.g., Montenbruck et al., 2018). However, the use of the float ambiguity solution in the BA processor is more than sufficient for generating high-quality RO products (e.g., Kursinski et al., 1997; Montenbruck et al., 2008; Innerkofler et al., 2020). The POD software embedded in the operational processor in 2021 was Bernese version 5.2. The OPE plot in Fig. 1 is the only based on data processed with Bernese version 5.2. This version has neither the non-gravitational force models nor the zero-differenceing ingeter-ambiguity-resolution algorithms available. However, as we argue below in this section, differences in the POD solution (due to e.g., the different models employed by the POD S/W), do not affect the BA statistics.

We assess the quality of a given POD solution in several complementary ways. There is an internal quality assessment within Bernese, where the final so-called reduced-dynamic orbit is compared to the kinematic orbit solution. A kinematic orbit

is obtained from the geometrical solution for the position of the LEO at the epochs in which observations are available (e.g., Švehla and Rothacher, 2005). A reduced-dynamic orbit indicates an orbit solution where the dynamical forces originating, e.g.,

from the extended gravitational field of the Earth, from the Sun, etc. are taken into account. The model cannot realistically take into account all possible forces, hence "reduced", and some empirical parameters compensate for this (see, e.g., Montenbruck et al., 2005, and references therein.). Bernese handles the data used to generate these two orbits in the same exact way (e.g., same data rejection parameters, same data weighting approach, etc.). As a result, the reduce-dynamics versus kinematic orbit comparison cannot detect any problem originating from, e.g., an error in the attitude modeling or in the definition of the position

of the POD antenna, since these kind of errors would affect the two orbits in the same exact way. Thus, we also use a different POD software, the NAvigation Package for Earth Observation Satellites (NAPEOS, Springer et al., 2011), to cross-check the consistency of the Bernese-based reduced-dynamic solution with the NAPEOS-based solution. As a result of being an altimetry reference mission, in addition to the RO POD antenna (RO-POD), S6A also features another POD antenna (GNSS-POD), which tracks both GPS and Galileo and is connected to a completely independent receiver with its own ultra-stable oscillator (e.g.,

Donlon et al., 2021). The comparison of the GNSS-POD solution with the RO-POD solution provides a receiver-independent way to cross-check the solution (of course the clock solution in this case cannot be compared, given that the two receivers have different oscillators). Furthermore, the GNSS-POD-based solution itself is routinely compared with solutions obtained by the members of the Copernicus POD quality working group (CPOD QWG) (Fernández et al., 2024), which EUMETSAT is part of. The above checks allow to perform several complementary S/W-, receiver-, and processing-center-independent cross

comparisons to validate our solution.

The focus of this work is to investigate the effect of different GNSS clock data rates on the RO BAs (cf. Fig. 1) and the effect is both direct, since they enter explicitly in the processing of each single RO profile, and indirect through their influence on the POD solution. In Fig. 3 we compare two S6A solutions for the month of November 2021 (which falls within our main test campaign, see Sect. 4.1). One is obtained using JPL RT GPS products, downsampling clocks to 30 seconds and orbits to

140 15 minutes, the other using CODE final products with clocks at 5 seconds and orbits at 15 minutes (sets JPL 30 s and COD 05 s in Table 1). The RINEX observation files, decoded from level zero data provided by the RO receiver at the rate of 1 s, are downsampled to 30 s when using JPL products and to 10 s when using CODE products. The use of 10 s RINEX data is the customary choice in POD reprocessing campaigns (see the discussion related to the regular service review in Fernández et al., 2024). However, in the POD processing, using a RINEX data rate higher than the clock data rate of the GNSS products would

not be beneficial, but would rather increase the noise of the solution (Dach et al., 2015). Thus, for the JPL 30 s, we use RINEX data downsampled to 30 s. It is clear from Fig. 3 that there is a good agreement between the two solutions, with biases below the millimeter level and RMS below 2 centimeters in any given direction. The comparison for the velocity solution (not shown) has both biases and RMS below 0.01 mm/s. These results are well within the standard target specifications of 5 cm and 0.05 mm/s for RO missions (e.g., Innerkofler et al., 2020).

As an additional example of the consistency of the POD results, we plot in Fig. 4 the three-dimensional root mean square (3D RMS) for a set of solutions, which includes the two cases of Fig. 3, based on data recorded by the RO-POD receiver, along with three solutions, based on data recorded by the GNSS-POD receiver, obtained by three members of the CPOD QWG. In

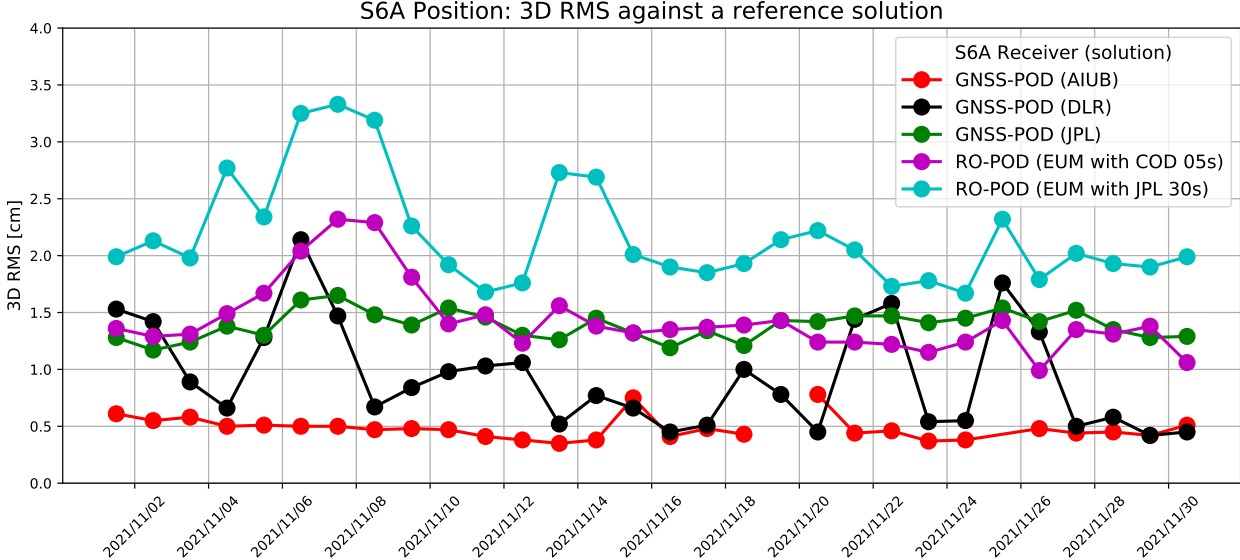

**Figure 4.** Comparison of different S6A POD solutions in the 3D-RMS space. The reference solution, against which the 3D RMS is computed, is the CPOD QWG combined solution (see main text for details). The pink and cyan solution are the same shown in Fig. 3, and are based on data of the RO-POD receiver, which tracks only GPS. The other three solutions are obtained by processing data of the GNSS-POD receiver, which tracks both GPS and Galileo. AIUB: Astronomical Institute of the University of Bern (their solution for Nov 19[th] was an outlier and has been removed); DLR: Deutsches Zentrum für Luft- und Raumfahrt; JPL: Jet Propulsion Laboratory.

Fig. 4, we use the CPOD QWG combined solution as reference solution (against which the 3D RMS is computed), which is a weighted average of the solutions provided by all the members of the CPOD QWG (see Sect. 3.6.2 in Fernández et al., 2024, for details). Typically, the spread of the different orbits in the 3D-RMS space is well below 3 cm. In Fig. 4 the only exception is the RO-POD solution obtained using JPL 30 s GNSS products (Table 1), which is expected given that it is the only set of data based on a Real-Time (and thus, less accurate) GNSS data stream.

### 3.2 GNSS auxiliary data

For the results presented in the following sections, the GNSS auxiliary data that we employ are JPL RT data (Table 1). The original orbit products are downsampled to 15 min. Different sets of GNSS auxiliary data are created by modifying the clock products. We use a straightforward downsampling (i.e., decimation) of the clocks to create four additional sets of clock products with rates of 2, 5, 10, and 30 seconds (JPL $dd$ s products, Table 1). Another obvious approach would be to fit the data of the 1 s product to obtain a smoothed version before downsampling it. We verified that for these products outliers are not a concern.

Furthermore, given that the choices that need to be made when fitting (e.g., function to use, degree of the polynomial in a polynomial fit, etc.) may have an effect on the results, we proceeded with the straightforward downsampling.

## 4 Results

Given the self-consistency of POD solutions obtained by using different GNSS products and approaches (Fig. 3 and 4), the focus of this section is on the effects of using GNSS clock products with different data rates in the BA data processing. The GNSS transmitter clock bias plays an important role when processing the reconstructed signal phases (i.e., the ones obtained by the level zero decoder) with a zero-differencing algorithm (Beyerle et al., 2005). The main test uses S6A data, and a validation test is performed using COSMIC-2.

For all the experiments presented in this section, the POD solution for the LEO has been obtained using the GPS products of the JPL 30 s data set (see Table 1) with clocks at 30 s and, correspondingly, RINEX data at 30 s (Sect. 3.1). For the RO processing each GNSS auxiliary data set has the same orbits and earth rotation parameters. Five different sets are used, corresponding to differences in the data rate of the clock products, namely 1, 2, 5, 10, and 30 seconds (JPL $dd$ s products, Table 1), as discussed in Sect. 3.2.

### 4.1 Sentinel 6A

We processed four months of S6A data, from September 1[st] to December 31[st], 2021, which corresponds to around 110k occultation profiles. The results on the bias and standard deviation against ECMWF-data-based forward-modelled profiles are presented per constellation.

For GLONASS, Fig. 5 shows that while the bias is unaffected by the use of different GLONASS clock data rates (in the 1 s–30 s range), the standard deviation markedly improves for impact heights above about 35 km. At 40 km the standard deviation decreases by about 1% going from a 30 second to a 1 second clock product. Using 30 s GLONASS clocks a 5% standard deviation is reached at an impact height of about 45 km, while with a 1 s clock this occurs at about 52 km.

Figure 6 shows that, as it is the case for the GLONASS occultations, the robust bias for the GPS occultations is practically unaffected by the different GPS clock data rates employed. Unlike GLONASS, though, the standard deviation only shows a minor dependence on the GPS clock data rate. However, as visible in the standard deviation at high impact heights between 50 and 60 km (Fig. 6, right panel), there is not a clear trend in the standard deviation with the increasing clock data rate, contrary to the case of GLONASS (Fig. 5). Given that the clock behaviour of the GPS constellation shows a block dependence (Fig. 2), we also plot the bias and standard deviation curves at high impact heights for the different GPS blocks separately (Fig. 7). Overall, for a given clock data rate, the standard deviation improves going from Block-IIR, to Block-IIF and to Block-III. However, for a given Block, the effect of increasing the GPS clock data rate does not affect the standard deviation curve consistently. For the Block-III (Fig. 7, right), going from 30 s to 1 s clocks brings a steady increase in the standard deviation at a given height, with the effect being really minor between 30 and 5 seconds, but becoming noticeable with 2 and 1 second clocks. For the Block-IIF (Fig. 7, middle), all sets are mostly coincident, with a slight degradation of the standard deviation when using the 1

**Figure 5.** Robust statistics of S6A GLONASS occultations against ECMWF-based short-range forecasts, for the period September to December 2021. Each set corresponds to bending angle profiles processed with different GLONASS clock data rates, as indicated in the legend (see also Table 1). Above an impact height of about 35 km, there is a clear advantage in using high rate GLONASS clocks.

second clock. For the Block-IIR (Fig. 7, left), increasing the clock data rate improves the standard deviation. The improvement is steady in going from 30 to 5 seconds. The 2 and 5 seconds curves coincides, while going to a rate of 1 second increases the standard deviation.

## 4.2 COSMIC-2

To verify the results observed using S6A data, we used data from entire COSMIC-2 fleet (e.g., Ho et al., 2020) in a smaller-scale test using three days in 2023 (August 5th–7th), which corresponds to about 9200 occultations. These results are based on processing of level zero data available on the UCAR server (UCAR COSMIC Program, 2019).

    As was the case for S6A, the increase in the clock data rate for GLONASS occultations improves the standard deviation with minor effects on the bias (Fig. 8). For GPS occultations we only present results for GPS clocks at 30 seconds (Fig. 9),

and these results for the standard deviation are consistent with what was found with S6A, where there is an improvement in going from Block-IIR, to Block-IIF, to Block-III (Fig. 9). Results for higher GPS clock rates are consistent with those obtained with S6A. The standard deviation for COSMIC-2 is worse than for S6A, despite having the same receiver. We attribute this difference to a POD solution of lesser quality: no macro-model has been used for COSMIC-2, the satellites orbit at a lower altitude (about 550 km versus the 1350 km for S6A) where atmospheric perturbations are larger, and the solar activity increased

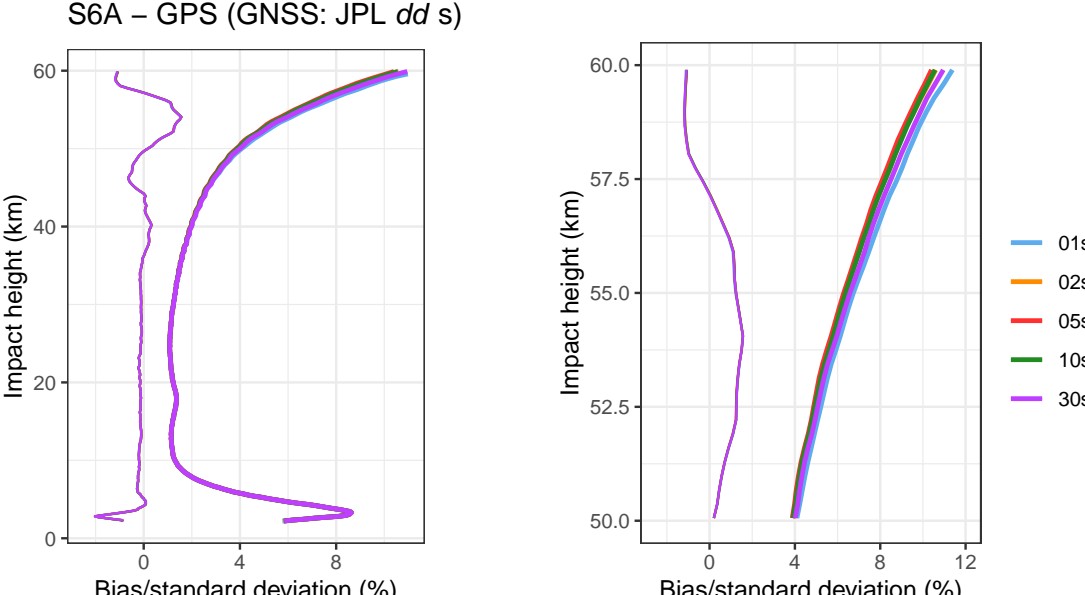

**Figure 6.** Robust statistics of S6A GPS occultations against ECMWF-based short-range forecasts, for the period September to December 2021. Each set corresponds to bending angle profiles processed with different GPS clock data rates, as indicated in the legend (see also Table 1). Unlike the case of S6A GLONASS occultations (Fig. 5), the standard deviation is largely unaffected by the GPS clock data rate used, as seen in the right panel, which is a zoomed-in view of the 50-60 km impact height range.

in 2023 with respect to 2021. Future work will further investigate possible improvements to the COSMIC-2 statistics through improved POD configurations.

## 5    Discussion

The analysis performed in this work points to the importance of using higher-rate GLONASS clock data to obtain the best bending angle statistical performances in terms of standard deviation at impact heights in excess of 35 km, with improvements

observed up to 1 s, which is the highest rate tested (Fig. 1, 5 and 8). There seems to be no simple dependence of the standard deviation curves for the GPS constellation as a function of the clock data rate (Fig. 7), and the 30 s clock products provide comparable performance to the GLONASS 1 s products (Fig. 1, 6 and 9). These results confirm expectations based on the analysis of GNSS clock noise at RO-relevant timescales (Hauschild et al., 2013; Griggs et al., 2015).

The above S6A results are based on global statistics of about 110k occultations. However, it is informative to also look at

single occultations to understand how the GNSS clock sampling rate affects the low-level processing of the measurements. We focus here on the zero-difference phase $P_{\mathrm{cr}}$ (linear units), which we define as the NCO (Numerically Controlled Oscillator)

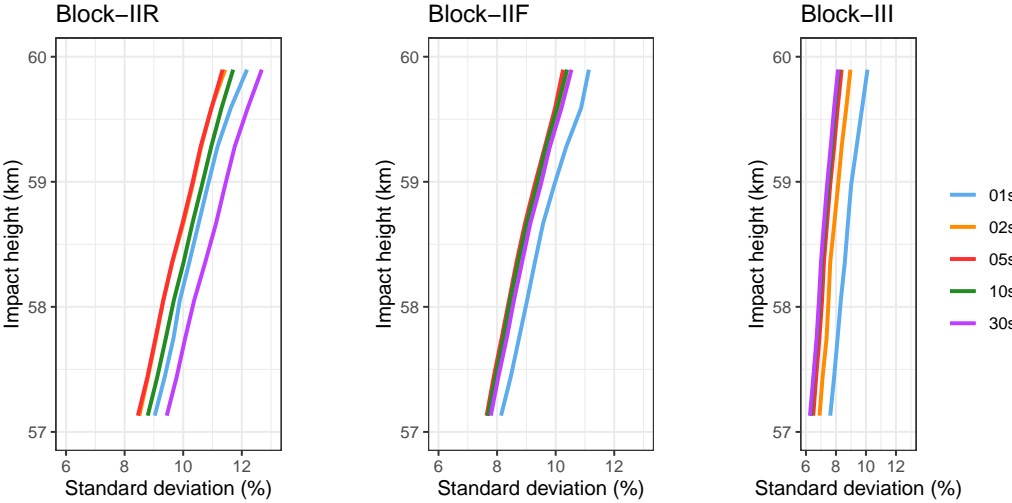

**Figure 7.** Above-50-km robust standard deviation of S6A GPS occultations against ECMWF-based short-range forecasts (as Fig. 6, right panel), for different GPS clock data rates (legend) and GPS transmitter types. Left: Block-IIR; Middle: Block-IIF; Right: Block-III. From left to right the standard deviation improves at any given clock data rate. However, the effect of increasing the GPS clocks data rate is not the same for each Block (i.e., panel).

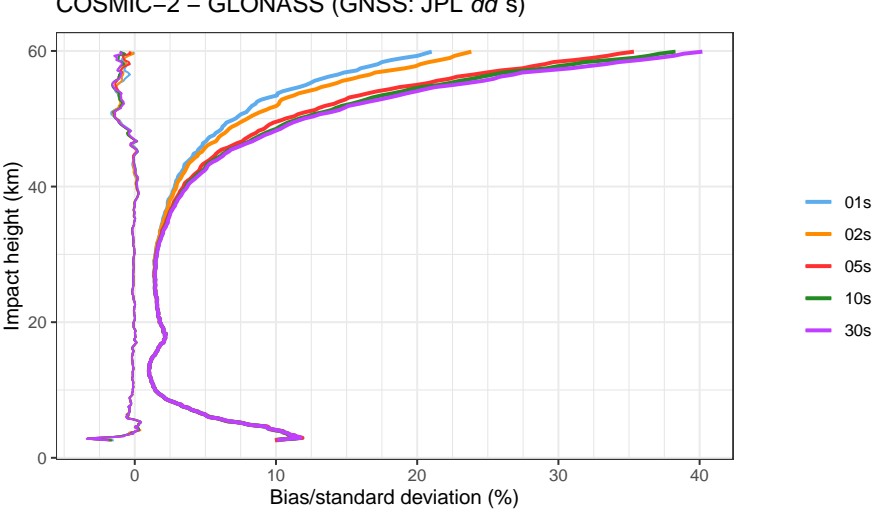

**Figure 8.** Robust statistics of COSMIC-2 GLONASS occultations against ECMWF-based short-range forecasts, for the period August 5th–7th, 2023. Each set corresponds to bending angle profiles processed with different GLONASS clock data rates, as indicated in the legend. Above an impact height of about 35 km, there is a clear advantage in using high rate GLONASS clocks, as was the case for S6A (Fig. 5).

**Figure 9.** Robust statistics of COSMIC-2 GPS occultations against ECMWF-based short-range forecasts, for the period August $5^{th}$–$7^{th}$, 2023. The curves have been obtained using 30 s GPS clock products. Each set corresponds to a different GPS Block, as indicated in the legend. As was the case for S6A for a given clock data rate (Fig. 7), Block-III is better than Block-IIF, which is in turn better than Block-IIR.

phase $P_{\text{NCO}}$ (cycles) corrected for the receiver ($\tau_{\text{rec}}$) and transmitter ($\tau_{\text{tran}}$) clock bias:

$$P_{\text{cr}} = c \left[ \frac{P_{\text{NCO}}}{f} - (\tau_{\text{rec}} - \tau_{\text{tran}}(\texttt{cr})) \right]. \tag{1}$$

In the equation, $f$ and $c$ are the frequency of the signal and the speed of light, respectively. Given that in our experiments the
LEO POD solution is always the same (and thus, $\tau_{\text{rec}}$ does not change), $P_{\text{cr}}$ can be used to assess the effect of the transmitter clock bias error on the measurements, since the bias error does depend on the clock rate $\texttt{cr}$ as indicated with the notation $\tau_{\text{tran}}(\texttt{cr})$. The quantity $\tau_{\text{tran}}(\texttt{cr})$ is the value of the transmitter clock bias linearly interpolated to the measurement times.

We calculate the difference of $P_{\text{cr}}$, for $\texttt{cr}$ equal to 2, 5, 10, and 30 seconds, with respect to $P_{\text{01s}}$, which we consider as reference. For the frequency L1, in Fig. 10 we plot the variation of the corrected phase $P_{\text{cr}}$ for each constellation and for each
hardware. We note the following:

1. GLONASS $P_{\text{cr}}$ variations (top row) are about a factor of three larger that the GPS $P_{\text{cr}}$ variations (bottom row);

2. For GLONASS, increasing the clock rate, significantly reduces the range of these differences;

3. For GPS, the reduction in the variations with increasing clock rate is minor, with the possible exception of the 30 s curve for Block IIR-A.

Similar features characterize the frequency L2 (not shown). Since the bending angle is proportional to the time variation of the phase (e.g., Kursinski et al., 1997), it is reasonable to conclude that what we see in these single-occultation-level analysis maps

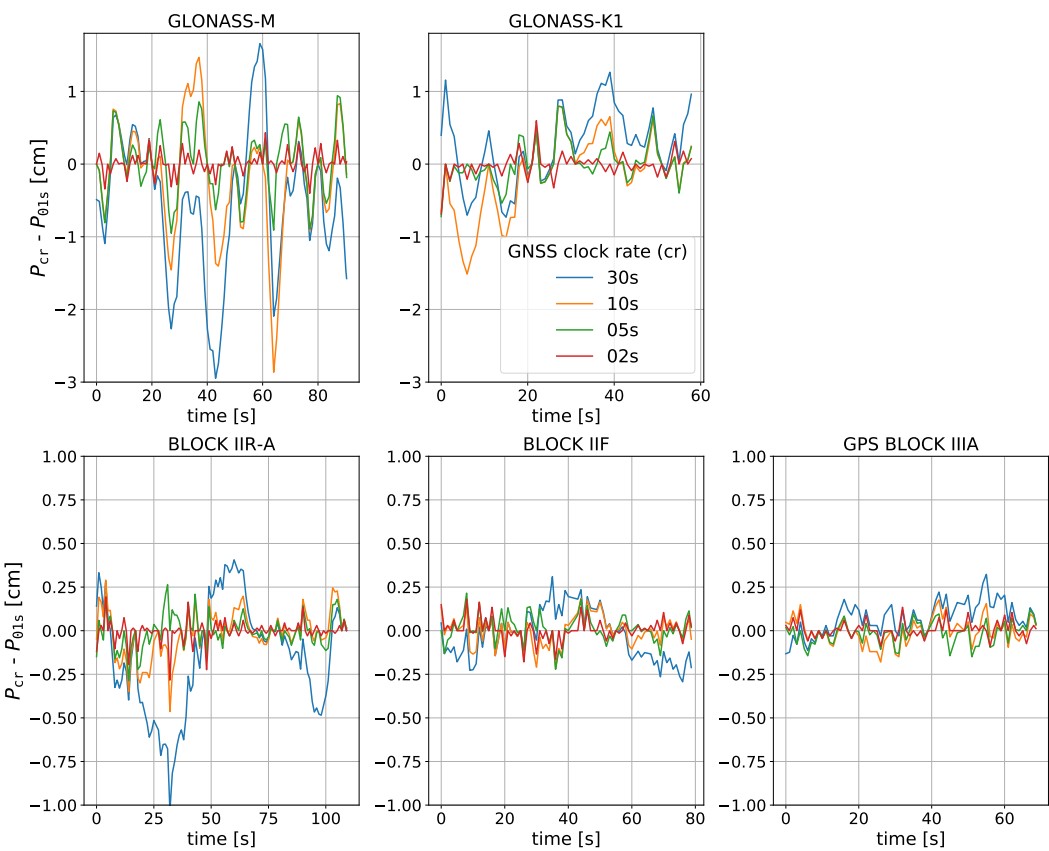

**Figure 10.** Variation of the L1 phase obtained with different clock rates $P_{cr}$ (legend) with respect to the phase obtained using 1 s clocks, $P_{01s}$. The panels refer to two GLONASS (top row) and three GPS (bottom row) setting occultations. (Note that the vertical axis range is different between the top and the bottom row.)

to the trend (or lack thereof) of the standard deviation curves in Fig. 5, 6, 7, 8, and 9. It is also consistent with the better Allan deviation shown by the GPS satellites (Fig. 2).

We also investigated the impact of the different GNSS clock rates on the vertical error correlation. The higher the vertical correlation is, the lower is the information content of a given profile (Bowler, 2020), with implications for the assimilation of these data in numerical weather prediction (NWP) models (e.g., Gilpin et al., 2019; Bowler, 2020). The results are shown in Fig. 11. With a behavior that is by now familiar, increasing the clock rate for GLONASS leads to a marked improvement of the vertical error correlation, while for GPS the variation is minimal. Once again, it makes sense that by sampling the noisier GLONASS clocks with a higher rate, leads to a less-correlated set of bending angle profiles. Finally, it is interesting to note that while there is no obvious trend in the standard deviation across the GPS blocks with respect to different clock rates (Fig. 7), in the vertical correlation plot the higher the clock rate, the better the vertical correlation, even though the improvement is limited.

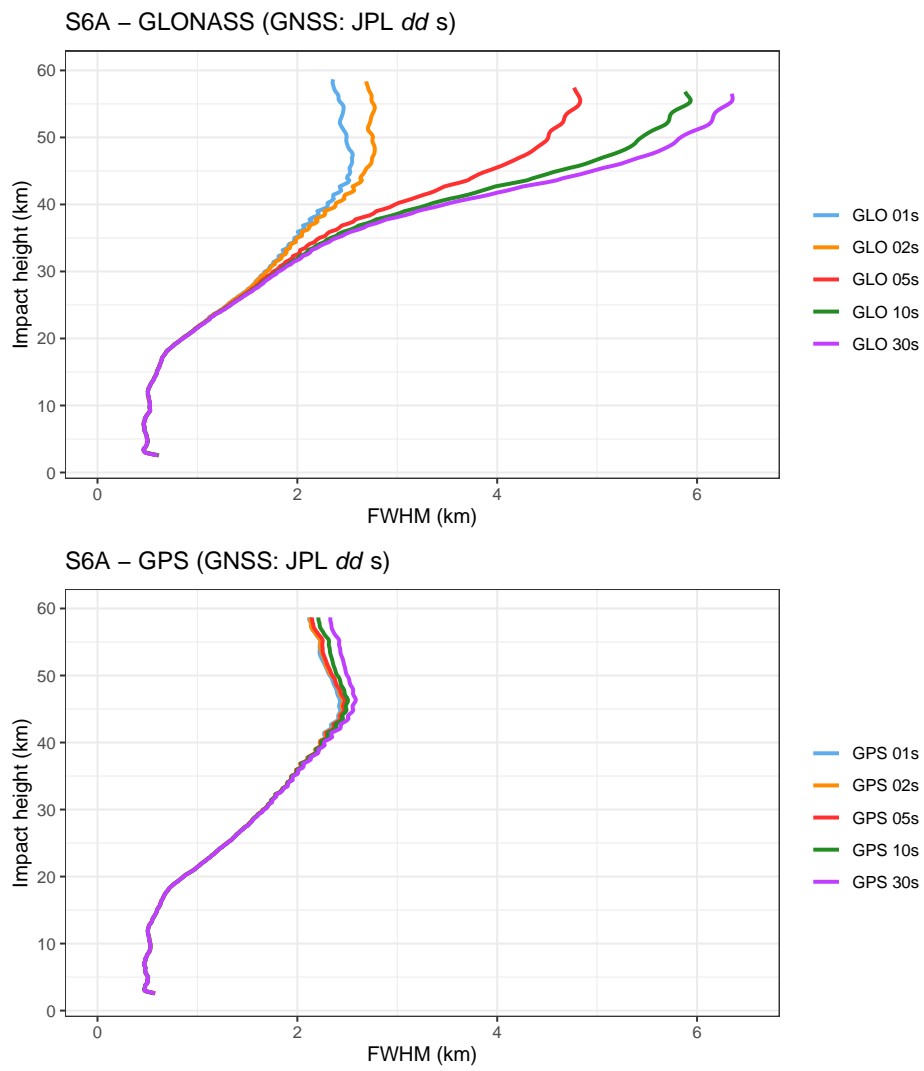

**Figure 11.** For S6A GLONASS (top) and GPS (bottom) occultations, vertical error correlation, shown here as the full width half maximum (FWHM) evaluated from the spread around the diagonal of the vertical correlation matrix at each level. The dataset used for this figure is the the same S6A dataset used in Fig. 5 and 6.

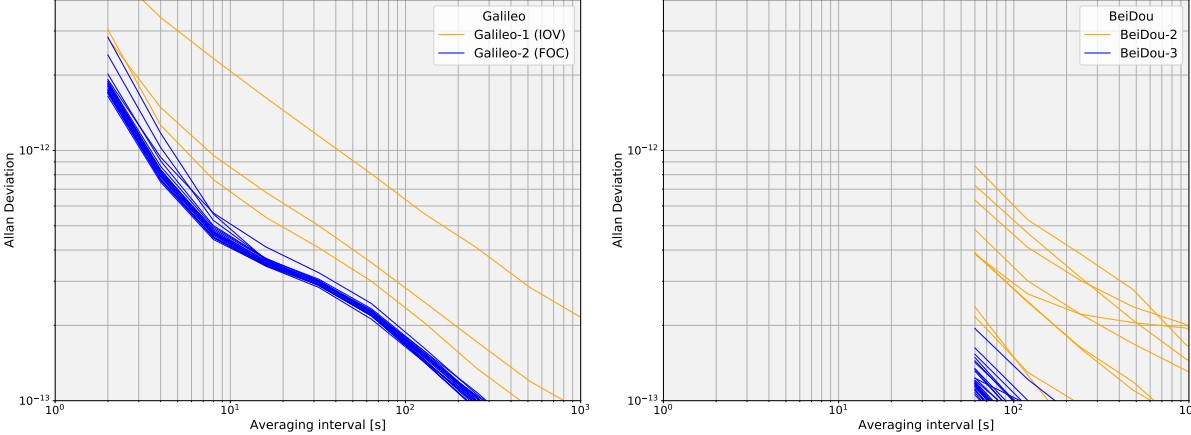

**Figure 12.** Modified Allan Deviation for December 31$^{\text{st}}$, 2023 for (left) the Galileo constellation, calculated using JPL RT Galileo clock data (available from the websites listed in note 5 of Table 1) and (right) the BeiDou constellation calculated using CODE MGEX data (available at http://ftp.aiub.unibe.ch/CODE_MGEX/CODE/).

The observed connection of the Allan deviation (i.e., the stability of a given clock hardware) with the standard deviation associated with the robust statistics of the BA profiles (Fig. 5 and 6), the phase variations in single occultations (Fig. 10) and the vertical error correlation (Fig. 11) points at the possibility of using the Allan deviation to guide our expectation for future analysis of Galileo and BeiDou datasets. In Fig. 12 (left) we show the modified Allan deviation for the 1s JPL RT clock products for the Galileo constellation on December 31$^{\text{st}}$, 2023. As with GPS (Fig. 2), the older satellites (In-Orbit Validation, IOV) have a higher Allan Deviation than the newer satellites (Full Operational Capability, FOC). Over the whole interval range shown in Fig. 12 (left), the values are comparable to those of the GPS constellation in Fig. 2. The JPL RT stream only provides GPS, GLONASS, and Galileo products. For BeiDou, we used the CODE data from the Multi-GNSS EXtentsion (MGEX, Prange et al., 2020), whose clock products have a rate of 30 seconds, and the Allan Deviation can only be computed for averaging intervals of 1 minute or longer (Fig. 12, right). In this limited region the values for the newer BeiDou-3 satellites are below those of GPS and Galileo. The curves in Fig. 2 and 12 are in line with recent estimates of the Allan Deviation of GNSS clock hardware (Jaduszliwer and Camparo, 2021). Based on these results, and under the assumptions that the BeiDou curves extend smoothly at short averaging intervals, we expect that using 30 s Galileo and BeiDou products will provide results similar to those we reported for GPS products in this study.

## 6 Conclusions

This work focused on the impact of using different GPS and GLONASS clock rates on the occultations recorded by the S6A and COSMIC-2 RO receivers. We evaluated the statistical properties of BA profiles obtained using GNSS clock products with different rates by comparisons with forward-modelled BA profiles based on ECMWF short-range forecasts. The use of higher-

rate clock products has almost no impact on the bias statistics, but it has an effect on the standard deviation above about 35 km. For GPS occultations, there is a marginal improvement in going from 30 s to 5 s, and overall more recent blocks show better standard deviation. For GLONASS, the improvement is very pronounced and continues for clock rates up to 1 s, which is the highest rate we tested (Fig. 5 and 6). Similar to the standard deviation, the vertical error correlation for GLONASS occultations markedly improves with the use of high-rate clock products, while the improvement is limited for GPS occultations (Fig. 11). These results are also reflected in the refractivity statistics derived from the Radio Occultation Meteorology Satellite Application Facility (Fig. 3.17 in Syndergaard and Lauritsen, 2021).

The results we obtained are a manifestation of the stability of the GNSS clock hardware that can be characterized through the Allan deviation, which is overall better for the GPS constellation with respect to the GLONASS constellation (Fig. 2). In the near future, the RO instrument onboard S6A and the one in the forthcoming follow-up mission, S6B (launch expected from late 2025 onwards), will start collecting Galileo signals on the RO antennas, in addition to GPS and GLONASS (EUMETSAT Public Document EUM/LEO-JASCS/DOC/21/1253471). This upgrade will provide a new dataset that can be used to extend the analysis performed here to include the Galileo constellation and test our expectations, based on the Allan deviation (Fig. 12). In addition, modern RO missions also exploit Galileo and BeiDou signals, as exemplified by the large dataset of the Radio Occultation Modeling Experiment (ROMEX, McHugh et al., 2023; Anthes et al., 2024).

For RO observations, the best combination of accuracy and small uncertainty is expected in the 5 to 30 km range (Kursinski et al., 1997), also referred to as the so-called "RO sweet-spot". Our results are consistent with this expectation in the accuracy (i.e., bias), but in terms of uncertainty (i.e., standard deviation), they show an increase below 10 km, as also reported in other studies (e.g., Anthes et al., 2022, and references therein). Recently, RO observations have been included as key observations in the assessment of temperature trends in the upper troposphere and lower stratosphere (Masson-Delmotte et al., 2021, Figure TS. 10, pg. 70). Given that GLONASS higher-rate clock data clearly improves the uncertainty and reduces the vertical error correlation, RO processing centers should aim at using this type of products, given their potential impact on both NWPs (e.g., Lonitz et al., 2021) and climate studies (e.g., Gleisner et al., 2022).

**Competing interests**

The contact author has declared that none of the authors has any competing interests.

**Acknowledgements**

We thank the two anonymous reviewers, whose comments helped improving this paper considerably. The research conducted by C.R.G. was carried out at the Jet Propulsion Laboratory, California Institute of Technology, under a contract with the National Aeronautics and Space Administration (80NM0018D0004).

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
