# Peer review of "Observed impact of the GNSS clock data rate on Radio Occultation bending angles for Sentinel-6A and COSMIC-2"

_Atmospheric Measurement Techniques, 2024_

## Author Comment (AC1)

General comments:

The paper describes the impact of using different clock rates in the correction/processing of GNSS observations and retrieval of bending angles (BA) from radio occultation (RO) data. It is shown that using increased clock rates for GLONASS occultations reduces the standard deviation in Sentinel-6 and COSMIC-2 BA relative to ECMWF forecasts at high altitudes. This is an important result for both NWP and climate applications. It is also shown that for GPS occultations, the residual clock noise mostly depends on the GPS transmitter type (and to a smaller extent the used clock rate), with newer GPS Blocks having less clock noise.

The paper is well organized and well written, and I recommend publication with minor revision. My comments below are mostly suggestions for text improvements. One suggestion (see below) goes to providing one example (one occultation) of the impact on the excess phase as a function of time for different clock rates. I think it would be interesting to see this and it would in my view improve the paper, but it is only a suggestion.

Dear reviewer, thank you for your feedback. We welcomed your suggestions and have looked at both the effect of the clock rate at the single-occultation level and on the vertical error correlation (see new Sec. 5). Please see below for a point-to-point response to your comments.

Below you'll find a point-by-point response to you comments.

Specific comments:

Line 40: I would add "and results are presented in section 4" after "set-up is described in section 3". I know it is implicit when you have parenthesis "(section 4.1)" and "(section 4.2)" in the next sentence, but it becomes more clear with an explicit statement. Also the parenthesis "(section 2)" in the previous sentence could be "as described in section 2" to be explicit. Normally parentheses should just contain clarifying information, not changing the meaning of the sentence if they were to be left out.

Text updated accordingly.

Line 58: "forward-modelled bending angle profiles extracted by the European Centre for Medium-Range Weather Forecasts (ECMWF) short-range forecasts" I suppose what is meant here is that the bending angles are forward-modelled based on temperature, pressure, and water vapor profiles extracted from ECMWF short-range forecasts. But that's not what the sentence says. The sentence should be revised.

Thanks, indeed that's what it's meant here, and we reworded accordingly, adding also a reference to the ROPP.

Line 78: slight -> slightly

Updated

Line 114: It is unclear to me what "processing-center-independent cross comparisons" mean. If it was the opposite, i.e., processing-center-dependent cross comparisons, I would maybe understand it as comparisons of solutions from different processing centers. However, strictly speaking you are saying that it is the comparisons that are processing-center-independent. I'm not sure if this is really what you want to say. I have the same issue regarding "SW-" and "receiver-" if the dash refers to "independent". There is also "receiver-independent way" in line 110 that I don't fully understand. Overall, could the sentences be re-formulated to become more clear?

You make a subtle point, but we think that the wording is correct: If the solutions we compared were based only on the GNSS-POD receiver, the result could be biased, if, e.g., there was any problem in the receiver itself. Once we compare the solution obtained with data from the GNSS-POD receiver with the one obtained with data from the RO-POD receiver, the comparison is then receiver-independent, in the sense that it does not rely on a single receiver. The same applies when using two independent S/W and when comparing solutions computed by different processing centres. We think that this should be clear.

Line 119: I would say "(cf. section 1)" if what you mean is that you already stated something similar. In section 1, the similar statement (line 20-22) refers to both positions and clocks, but here only to clocks. Maybe the reference to section 1 here is not needed at all.

We agree, the reference to section 1 has been removed.

Line 120: "One is obtained using JPL RT GPS products with clocks at 30 seconds and orbits at 15 minutes". I suppose you here downsample to 30 seconds and 15 minutes, respectively. I think you should include that in the sentence, since these numbers are different from those given in the introduction. E.g., ... using JPL RT GPS products, downsampling clocks to 30 seconds and orbits to 15 minutes".

Thank you for noticing this. We changed the text accordingly. We also added Table 1 to summarize all the GNSS data sets that we used.

Line 136: fit -> smooth (I suppose that is what you mean).

We'd use a fit to smooth the data. The text has been reworded accordingly.

Line 136: Not sure if this is necessary: "For example, if every 30th data point were an outlier, the 30 s downsampled product would contain little useful information. With this caveat in mind, the direct decimation represents a conservative choice since it could increase the relative percentage of outliers, thus adding to the loss of information in the downsampled products." It seems obvious that this could in principle happen, but is it a real concern?  Is there any reason to believe that the fraction of outliers would increase? If there is no real concern, I think these sentences could be removed.

Thank you, indeed it was not necessary to include this sentence. We removed it and slightly reworded the text.

Line 147: a RINEX -> RINEX data

Updated.

Line 148: Also here "(cf. Section 3.1)". Should it be "Section" or "section"? I'm not sure what the AMT guideline says.

Thank you, indeed the guidelines indicate that "Sect." should be used (and also, that for figures "Fig." should be used). We updated the entire text accordingly.

Line 149: illustrated -> mentioned.

Updated with 'discussed'.

Section 4, before Section 4.1: Would it make sense to give one example of excess phase as a function of time over about half a minute at the beginning of a setting occultation where the excess phase is close to zero (with not too much ionosphere) to show how the noise in excess phase is reduced when the clock rate increases? I haven't seen that before, and I think it would

be very illustrative. It would help to understand the characteristics of clock errors (and corrections) on the phase data. Excess phase differences using two clock solutions at different rates might also be very illustrative - I have never seen it.

Thank you for this suggestion. We now have a separate "Discussion" section, where we show plots that illustrate the impact of the GNSS clock rate on both the single-occultation level and on the overall vertical error correlation.

Line 153: "following the approach of Figure 1". Do you mean "similar to Figure 1"? Maybe it is not needed in the sentence.

Reworded.

Figure 5: It is difficult to distinguish between 01s and 30s, as well as between 02s, 05s and 10s (also in the right panel). Maybe choose more distinguishable colors.

The figures have been improved in quality and the color palette has been updated for better readability.

Comment to the results in Figure 5: Although there is not much gained in terms of standard deviation using higher clock rates for GPS, there could be a difference in error correlations. This in turn could affect standard deviation in derived refractivity (or in data assimilation of bending angle). I'm not asking for refractivity statistics here, but maybe my suggestion above of giving an example in excess phase could shed some light on this.

Thank you for this suggestion. We now have a separate "Discussion" section, where we show plots that illustrate the impact of the the GNSS clock rate on both the single occultation and on the overall vertical error correlation.

Figure 6 caption: hardware -> GPS transmitter types (I suppose), not unique -> not consistent with the other GPS Blocks (or something similar).

Reworded.

Figs 4,5,6,7.8: Discussions of results in the last sentences of the figure captions should be moved to the text (or just removed if already discussed in the text).

We think that a figure caption that also delivers, concisely, the main points, is worth having to help the casual paper skimmer, even though this results in some redundancy between text and captions.

Line 177: "there is an improvement in going from Block-IIR, to Block-IIF, to Block-III". But is the picture for a given Block about the same for COSMIC-2 as for S6A?

No, for C2 the statistics are worse. The reason is a lower quality of the POD solution for the C2 spacecrafts. A sentence has been added to the main text.

Line 184: I couldn't find anything about clock noise in (Harnisch et al., 2013). Please check if it is the right citation/reference.

Thanks, it should have been indeed Hauschild et al. 2013. Fixed.

Line 190: Maybe the part on ROMEX needs update, or could a more general statement be made?

This part has been partly rewritten, and the ROMEX experiment is now referred to only as a dataset comprising all four large GNSS constellations. We don't mention it directly as a possible source of data for studies like this one, since most of the ROMEX data is at the L1A level.

Line 194: I couldn't find the word "sweet-spot" in (Kursinski et al., 1997). Maybe this citation is not needed here.
Indeed it's an informal definition, we move the reference to avoid confusion.

---

## Author Comment (AC2)

**Summary**

The present paper discusses the impact of different rates (1 to 30 seconds) from the GPS and GLONASS constellation clocks on radio occultation (RO) bending angle profile statistics for Sentinel-6A and COSMIC-2 RO missions. The authors conclude that higher GNSS clock rates lead to improved bending angle profile statistics with decreased standard deviation in comparison to ECMWF short-range forecast data for GLONASS, while not much is gained by using higher rates than 30 seconds for GPS (with slight differences between different GPS blocks). This is supposedly related to known lower short-term stability of GLONASS clocks.

**General comments**

Admittedly, the findings in this paper are not entirely new and the authors should include previous work conducted with regard to this topic in their discussion even if from preliminary studies, e.g., Yao et al. 2023 (also reference [1] therein) who investigated the effect of higher-rate GPS and GLONASS clocks with respect to COSMIC-2 in a similar study setup. However, the work presented discusses the issue in more detail than previous studies (at least to my knowledge, the authors are advised to conduct a proper literature review for other related publications) and the authors present the results in a well-structured way providing extensive illustration supporting their argumentation.

In order to strengthen and enhance the leverage of the publication the authors should therefore expand their discussion for the following points. Since the focus of the study is on the impact of GNSS clock rates on RO data processing, it would be of interest to discuss and show how the different clock rates are applied and manifest in the RO excess phase calculation, as the point where GNSS clock data enter the RO processing. Additionally, the applied interpolation method from the GNSS clocks to the high-rate RO measurement time stamps should be included in the paper and possible implications of different interpolation methods, if there are multiple to choose from, should be discussed (in dependence of the clock rate, if relevant.

Yao, Jian, Weiss, Jan-Peter, VanHove, Teresa, "Impacts of High Rate GNSS Satellite Clock Estimation on Radio Occultation Bending Angle Retrievals: Preliminary Report," Proceedings of the 2023 International Technical Meeting of The Institute of Navigation, Long Beach, California, January 2023, pp. 995-1001. https://doi.org/10.33012/2023.18621

Dear reviewer, thank you for your feedback. Both you and the other reviewer suggested to have a look at the effects of the clock rate at the single-occultation level. This analysis is now included in Section 5, where we also investigated how the clock rate affects the vertical error correlation.

For the clock we use a linear interpolation, and this information is also included in Section 5.

We added a reference to Yao's extended abstract.

**Line per line and figure specific comments:**

Figures (general): On my printout the graphics are slightly blurred, please provide the figures in higher resolution. Use intermediate minor tick-marks and provide major tick-marks with shorter intervals to support the reader with the identification of relevant values in figures (applies basically to all figures except Figure 2).

Figures have been updated for quality and clarity.

L2: "Space-based RO experiments …". For my understanding this sounds a bit too "experimental", RO is a proven and well advanced remote-sensing measurement technique, but maybe this is commonly recognized designation. This applies to other occurrences in the text as well.

We just had in mind RO experiments on other planets, where the receiver sits on ground and not on a LEO. But indeed in the context of this special issue, there's no need to specify it.

L2: What is meant by RO experiments "currently" require tracking of signals from GNSS? I suggest to remove "currently".
We had in mind the proposed LEO-LEO RO experiments. But given that no such experiment exists at the moment, we removed "currently".

L5-6: Radio occultation was already introduced as acronym, use RO acronym here.

Done.

L6: Update to "…, where the orbit and clock information for the transmitter (GNSS) and receiver (LEO) satellites is required."

Updated.

L9: "… the study focused on the effect", instead of "…, the focus will be on the effect". Remove coma after "data rates".

Updated.

L10: "… range from 1 to 30 seconds". State which exact rates were applied if this can be stated generally. This is more informative.

Reworded.

L11: State which four month served as test data period for Sentinel-6A and also which dates served as input for the COSMIC-2 analysis.

Done.

L13: Depending on the context 30 seconds can already be considered high-rate compared to e.g., 5 or 15 minutes orbit sampling. Better to use "… higher-rate clock information". This might apply to other occurrences in the text as well.

Done here.

L17: I suggest to replace "estimation" by "calculation" and rephrase the following part "… BA profiles based on signals from …".

Done.

L18: Remove "an" in "requires an accurate".

Done.

L25-26: Unclear language. Rephrase to something like: "Due to the random stochastic noises that affect GNSS atomic clocks, a smaller sampling interval is required to obtain accurate interpolations, …"

Done.

L29: Aren't the CODE final GNSS orbits provided with 15 min sampling? Please check.

They moved to 5min since the transition to IGS20. We included this info in the GNSS Table.

L35-37: This sentence does not seem logical, please state clearly what you are trying to say. Are you saying these stability analyses are "used" for high-rate corrections in order to obtain high-quality BA products. If yes, in which way are they used?

Reworded.

L42: Add verb. "The discussion and conclusions are presented in Section 5". Makes sure to follow the journal guidelines for the upper/lower case notation of keywords like Section, Figure, etc. and follow them consistently throughout the manuscript.

Done.

L45: Better to use past tense: "... was equipped ... was built ...".

Changed to "is equipped" (since S/C is still flying) and "was built" as suggested.

Figure 1: Please include OPE and STC in the title of the two figure panels, respectively. This way the reader has a direct connection to the acronyms used in the text. Also add it in "Left: Operational data stream (OPE) ..."

Done.

L58: Please correct: "... bending angle profiles are extracted from ...".

Reworded.

L62: Please indicate if you are you using matching occultation events for both processors or if they differ.

Added a sentence to clarify.

L64-67: You mention JPL final clocks used for OPE for GPS. Where have the JPL final products been introduced? I can't find them in the paragraph from L23-32. Also, why are you using different inputs for OPE: GPS (JPL final 30 s) and GLONASS (JPL RT 1 s)? In my view it would be clearer and more consistent to use the same input for both, GPS and GLONASS. Please clarify.

Clarified, and a table has also been added, to identify the rates of each set of products.

L71: Did you look for more up to date references? I did not check but Jaduszliwer et al. 2021 and references therein might be a starting point.

Jaduszliwer, B., Camparo, J. Past, present and future of atomic clocks for GNSS. *GPS Solut.* *25* , 27 (2021). https://doi.org/10.1007/s10291-020-01059-x

Thank you for this suggestion. We included this reference when we discuss the AD of the Galielo and BeiDou constellations in Sec. 6.

L80: Rephrase "... described in the following are based on the re-processing of RO data ...".

Done.

L81: Please provide more details which RO receiver type is flown on S6A and COSMIC-2 and provide a reference.

Done.

L84: Is there any reference for the YAROS software?

Unfortunately not.

Figure 3: Please add a grid and intermediate tick-marks on the y-axis to support the viewer. Add "Sentinel-6A" to the title.

Done.

L96-98: So BSW5.2 was used for OPE processing and BSW5.4 for STC processing? Please clarify.

Reworded for clarity.

L103: Remove punctuation: "See, e.g, …".

Done.

L104: Plural: "… the reduced-dynamic and the kinematic orbits …".

Done.

L104: What exactly do you mean by the data handling component? The same software using the same processing setup? If so, please revise to improve clarity.

We reworded and expanded this segment.

L112: What are the different characteristics of the two oscillators?

The point here is that the measurements are time-tagged differently, since the oscillators are different. So, independent of the characteristics of these oscillators, their solution cannot be directly compared.

L115: You state that typically the spread of the different orbit solutions is below 3 cm 3D-RMS. Did you check this for the investigated time periods in this paper as well? Did you find any noteworthy deviations? It would be a valuable addition for the reader and further improve the manuscript to add a time-series plot of the 3D-RMS of the different comparisons for Sentinel-6A and the same time period as in Figure 3.

We now include an additional figure with the 3D-RMS for a set of solutions (Fig. 4).

L117: Better: "… (cf. Figure 1) …".

Done.

L120: In the paragraph from L34-32 you introduce JPL RT products with 1 min orbit and 1 s clock sampling but here you are using JPL RT products with 15 min orbit and 30 s clock sampling? Also, since it is hard to keep track which GNSS product is provided with which orbit and clock sampling rates one could consider to collect this information in a concise table at the beginning of the manuscript.

Thanks for this suggestion. We now added a Table collecting the information on the various set of GNSS products used, and refer to it where necessary.

L124: This has been shown by Fernandez et al., 2024? If yes, make this more clear by saying "… it has been shown by Fernandez et al 2024 that not much is gained in this case".

The use of 10s RINEX data is standard practice in the POD community and it's the common choice among the CPOD quality working group members when obtaining solutions for altimetry

missions (which have the most stringent POD requirements). In Fernandez et al., 2024 this is not explicitly indicated, so we reworded the sentence.

L127: Provide proper figure reference: "It is clear from Figure 3 that …"

Done.

L133: Again I am confused by 15 min orbit sampling here and the 1 min orbit sampling of JPL RT data at the beginning of the paper.

We clarified and added a reference to the Table.

L135-139: You state that using a fit would be more stable against outliers and therefore the better option, still you use straightforward decimation. What is the estimated difference between the two approaches and can you confirm that your choice does not impair the results?

We added some information, indicating that outliers are not a concern for these products, and the reason for proceeding with the simple decimation.

L141: "… obtained by"

Done.

L146: Doesn't Sentinel-6A track GPS and Galileo on the POD antenna?

Only for the GNSS receiver. The RO receiver only tracks GPS in the POD antenna. We clarified this in the text.

L149: Better: "… as discussed in Section 3.2."

Done.

Figure 4: Please include the time period of the underlying data in the figure caption. Also in Figure 5, 7, and 8.

Done.

L160: Add the height range to the text for convenience: "… at high impact heights between 50 and 60 km"

Done.

L176: You limit the illustration of COSMIC-2 GPS occultations to 30 s clock products in Figure 8. For consistency and to underline your findings it would be interesting to include the full range from 1 s to 30 s clock rates, in the way how it was done for Sentinel-6A or at least state that the COSMIC-2 analysis shows similar characteristics if this is true.

We now explicitly mention that the results are similar.

L180: Best performance in terms of standard deviation of what? Remind the reader that your analysis is based on bending angle statistics here.

Done.

L186: What is meant by "real" RO observations, are there any other? I suggest to remove "real".

Done.

Figure 6: Why does the standard deviation for different clock rates reach different impact heights? E.g., in the left panel (Block-IIR) the green line (5 s) reaches about 58.7 km while the blue line (10 s) ends at 58 km?

The figure (which is now Fig.7) has been updated.

L191: It would have completed the picture to include Galileo and Beidou occultations in this study. Nevertheless, do you have any expectation on the behavior of the use of higher rate clock data from those constellations and their impact on RO bending angle statistics, given their clock stability?

We added a figure for the AD of Galileo and BeiDou in Sec. 6, and included a discussion of their expected performance in RO BA statistics.

L193-195: You mention the RO sweet-spot down to 5 km, but what about the increased standard deviation shown in your figures at these altitudes?

Indeed, we reworded to point out that the standard deviation is worse below 10 km.

L202: Acknowledgements

Done.

L221: Remove repetition of "https://doi.org/" and what is "112 395", seems odd.

Done.

L223: Remove repetition of "https://doi.org/".

Done.

L241: Remove repetition of "https://doi.org/".

Done.

L257: Remove repetition of "https://doi.org/".

Done.

L259: Add DOI.

Done.

L266: Add DOI. This is still in "Atmos. Meas. Tech. Discuss.", update to published paper if available.

Done.

L273: Is there a online resource available?

Yes, added.

L275: Remove redundant "2019" before DOI.

Done (it was the volume number, but wrong).

---

## Referee Report (RR1)

**Observed impact of the GNSS clock data rate on Radio Occultation bending angles for Sentinel-6A and COSMIC-2**

Sebastiano Padovan, Axel Von Engeln, Saverio Paolella, Yago Andres, Chad R. Galley, Riccardo Notarpietro, Veronica Rivas Boscán, Francisco Sancho, Francisco Martin Alemany, Nicolas Morew, and Christian Marquardt

https://amt.copernicus.org/preprints/amt-2024-80/

**Summary**

The authors provided a revised version of the manuscript addressing the comments from the previous review. One major improvement is the inclusion of an analysis investigating the effects of different GNSS clock rates at single-occultation level. Additionally, Section 3.1, which covers precise orbit determination of the Sentinel-6A (S6A) satellite, was strengthened with a comprehensive comparison of different orbit solutions presented in Figure 4. To provide a better overview in support of the reader, the authors added a table summarizing the different sets of GNSS products used in the study. Furthermore, the final combined Discussion and Conclusions section was separated and expanded, now also including considerations on the Galileo and BeiDou GNSS systems.

**General comments**

In response to feedback from the initial review, the authors included an additional figure and corresponding discussion regarding the expected performance of Galileo and BeiDou occultations in the Conclusions section. While the addition of this content and the insights on these GNSS systems is valuable and enriches the publication, the introduction of a new figure and content in this final manuscript section is not common practise. It is required that the authors restructure the last two sections of the manuscript and move their main considerations on Galileo and BeiDou, along with Figure 12, to the Discussion section. Since the final Conclusions section is generally intended to be self-contained, it is further recommended to minimize figure references unless they are considered essential.

The paper primarily focuses on S6A and utilizes only a small batch of COSMIC-2 data. Section 4.2 points out a larger standard deviation in the bending angle statistics for COSMIC-2 compared to S6A, attributed to a POD solution of lower quality. It is argued that this is caused by the absence of a satellite macro-model, the lower orbit altiude, and increased solar activity in 2023. While Section 3.1 extensively discusses S6A POD, there is a lack of discussion on COSMIC-2 POD in this dedicated POD section. It is advised that the authors add a short discussion of COSMIC-2 POD, including relevant numbers or references, in order to provide a complete analysis and to support their assessment of lower COSMIC-2 POD quality in Section 4.2.

**Line per line and figure specific comments:**

Figures (general): Please use intermediate minor tick-marks and provide major tick-marks with shorter intervals to support the reader with the identification of relevant values in the figures (applies basically to all bending angle statistics figures).

--- Abstract

L2: In the first review I noted the following: *"Space-based RO experiments …". For my understanding this sounds a bit too "experimental", RO is a proven and well advanced remote-sensing measurement technique, but maybe this is commonly recognized designation.*

We just had in mind RO experiments on other planets, where the receiver sits on ground and not on a LEO. But indeed in the context of this special issue, there's no need to specify it.

What I was referring to in my comment above was not "space-based" but the wording "experiments". I suggest to replace it by "measurements". Note that this applies to other occurrences in the text as well.

--- 1 Introduction

L20: Remove "an" in "requires an accurate knowledge ...".

L33: The wording "some information" gives the impression that Table 1 is somehow incomplete. I suggest to rephrase.

--- 2 Motivation

L59: Please introduce OPE at its first usage in the text, independently from Figure 1. In Figure 1 be consistent with the introduction of OPE and STC.

L65: Start new paragraph with: "The comparison of the statistics …".

L68: Note the mission: "…, using recent S6A data …".

L70: You state that different POD SW and different GNSS auxiliary data are responsible for a 3 % difference in total number of BA profiles obtained by the OPE and STC processors. The OPE uses GNSS products from JPL-EUM (GPS: 15 min/30 s; GLONASS: 15 min/1 s) and the STC GNSS products from COR (GPS: 15 min/30 s; GLONASS: 15 min/30 s). I wonder how the different POD SW and GNSS auxiliary data (used by JPL and CODE) affect the number of processed BA. I assume both analysis center deliver GNSS orbit and clock data products covering the entire test period and that for each retrieved profile a matching modeled profile exists. Is it rather the different quality of the provided orbit and clock data which differs and therefore leads to rejection or failure to process some of the occultations? Or are there differences between the OPE and STC processors leading to different numbers of successfully processed BA profiles?

L72: Please stick with OPE here instead with NTC. It is easier to relate to the following text and occurrences of OPE therein.

L76: Start new paragraph with: "Previous work has pointed out …".

L79: Plural and hyphenation: "Less stable clocks would require a higher-rate correction to compensate their noise, …".

---- 3 Set-up of the experiments

L109: At the beginning of Section 3.1 you state that LEO orbit and clock solutions are obtained with Bernese 5.4, with the exemption that the OPE embedded Bernese 5.2 in 2021 and thus also has been used with the OPE processor in Fig. 1 (L109). To clarify, apart from the OPE analysis in Fig. 1, which used Bernese 5.2, in all other cases Bernese 5.4 was used?

L106: Better: "Integer-ambiguity fixing has been shown to perform better …".

L120: "etc." instead of "ect."

L120: Please correct: "As a result, the reduced-dynamic versus kinematic orbit comparison …".

L124: Introduce hyphenation for "cross-check". Also for all further occurrences (e.g., L128).

L150: Sentence structure: "In Fig. 4, we use the CPOD QWG combined solution as the reference solution... ".

L153: Add missing space between value and unit: "JPL 30 s GNSS products".

--- 4 Results

L165: Plural: "GNSS clock data rates". Maybe even better to rephrase: "… using GNSS clock products with different data rates in the BA data processing".

L172: Remind the reader once more of the selected data rates and specify the five data sets used.

L197: Do the three-days of COSMIC-2 test data comprise data from all flight models and have they been processed from level 0 data from UCAR? Please add this information, in particular the origin of the data, to the manuscript.

L199: Sentence structure: "For GPS occultations we only present results for GPS clocks at 30 seconds (Fig. 9), …".

--- 5 Discussion

L206: Better: "… points to the importance …".

L207: Please correct to "… in terms of …".

L212: Use numerals rather than spelling out numbers (e.g., about 110k occultations).

L234: Plural: "these data".

L237: Plural: "… set of bending angle profiles."

L238: I think you intend to refer to Fig. 7 here. Also, I would rather say that there is no obvious trend in the standard deviation across the GPS blocks with respect to different clock rates.

Figure 10: I suggest the following adaptations to the figure caption: Move second part of the first sentence to the beginning of the sentence to improve the word order; Move the note on the different

vertical axis ranges in brackets to a separate sentence. Furthermore, in my opinion the interpretation rather belongs in the text than in the figure caption. In any case you should elaborate which decrease is very evident for GLONASS and rather talk about phase variations or differences instead of lines for GPS.

--- 6 Conclusions

Please see the general comments section for remarks on the content and structure of the Conclusions section. Besides that, line per line comments follow.

L241: Remind the reader once more what was the main focus of the study: "This work focused on the implications of different GPS and GLONASS clock rates on occultations recorded by …".

L243: Word order: "… S6B, will start collecting Galileo signals on the RO antennas, in addition to GPS and GLONASS signals."

L245: Plural: "… modern RO missions also exploit … ".

L248: Standard deviation of what? Elaborate and provide more context.

L249: "vertical error correlation (Fig. 11)"

L253: Add figure number and adequate commas.

L256: So there are no BeiDou clock products available with a rate higher than 30 seconds? Especially the shorter averaging intervals of the AD would be of interest.

L262: As you pointed out in your response to the initial review I also consider "sweet-spot" an informal definition because it varies among different studies. I recommend not overemphasizing it: for example, you may consider stating "...is expected in the 5 to 30 km range (Kursinski et al., 1997), also referred to as the so-called 'RO sweet-spot'.", and avoid the repetition of the term in the following sentence.

L264: Remove comma before "Recently …".

--- References

L298: Remove space from article number: "112395".

---

## Author Response (AR2)

I'm satisfied with the answers to my questions and improvements to the manuscript. I have a few comments and suggestions to the new parts of the manuscript, and a few very minor things that I didn't catch in my first review.

Thank you for these additional comments and please see below the point-by-point responses.

Line 13: "bending angles statistics" should perhaps be "bending angle statistics".

**Corrected.**

Line 14: I think the sentence would be better if "and" is replaced by "but".

**Agreed, it's indeed better.**

Line 15: impact-height -> impact height.

**We keep it hyphenated since it's a compound adjective.**

Fig. 3 caption: "CODE 05 s (5 min orbits and 5 s clocks).". Should it be 15 min (it says so in the text, line 137)

Yes, thanks for catching this one. It's now corrected.

Line 151: There is a "the" too much in "by the all the members".

Removed.

Fig. 7: "Bias/standard deviation" in the x-lables, but it is only standard deviation here.

Corrected.

Equation after line 215: Although it is the only equation in the paper, it should still have a number (1). Perhaps say that P\_cr is in terms of meters (as opposed to P\_NCO).

**Added the number and the units.**

Line 218: direct handle -> view. I don't understand 'handle' in this context.

**Reworded.**

Line 219-220: bias -> error. Bias is a statistical measure when averaging many observations, which is not what you show here (in Fig. 10).

For the transmitter/receiver clock solutions, we refer to a bias, since they indicate a (timevarying) offset with respect to the GPS time. However, to address your comment we now explicitly write "bias error".

Line 223: Some things are worth noting in the figure -> We note the following

**Reworded.**

Line 228: Being the bending angle proportional -> Since the bending angle is proportional

**Reworded.**

Line 249: (11) -> (Fig. 11)

Corrected.

I suggest moving the new discussion in section 6 about the Allan deviation to the discussion section (section 5).

Done.

Discussion section: Why not consider the use of 5s clocks for GPS in the future? It seems that 5s is good for all GPS blocks and also gives the smallest standard deviation in the overall results in Fig. 6. I think the results in Fig. 6 is in accordance with results of refractivity statistics in the ROM SAF report here: https://rom-saf.eumetsat.int/product\_documents/romsaf\_vr\_atm\_ntc.pdf, where it appears that the 5s GPS clock results are slightly better than those using 30s clocks at high altitudes (Fig. 3.17 in the report). However, both are worse than the 1s GLONASS clocks, perhaps due to the vertical correlations that increase standard deviation in refractivity.

Thank you, that's a good point and we added this remark in the abstract and conclusion section. Thank you also for the pointing us at the SAF report, which we now quote in the conclusion section. section.

**Summary**

The authors provided a revised version of the manuscript addressing the comments from the previous review. One major improvement is the inclusion of an analysis investigating the effects of different GNSS clock rates at single-occultation level. Additionally, Section 3.1, which covers precise orbit determination of the Sentinel-6A (S6A) satellite, was strengthened with a comprehensive comparison of different orbit solutions presented in Figure 4. To provide a better overview in support of the reader, the authors added a table summarizing the different sets of GNSS products used in the study.

Furthermore, the final combined Discussion and Conclusions section was separated and expanded, now also including considerations on the Galileo and BeiDou GNSS systems.

**Thank you for your additional feedback, and please see below for point-by-point responses.**

**General comments**

In response to feedback from the initial review, the authors included an additional figure and corresponding discussion regarding the expected performance of Galileo and BeiDou occultations in the Conclusions section. While the addition of this content and the insights on these GNSS systems is valuable and enriches the publication, the introduction of a new figure and content in this final manuscript section is not common practise. It is required that the authors restructure the last two sections of the manuscript and move their main considerations on Galileo and BeiDou, along with Figure 12, to the Discussion section. Since the final Conclusions section is generally intended to be self-contained, it is further recommended to minimize figure references unless they are considered essential.

We restructured the Discussion and Conclusions section, as suggested. We reduced the number of Figure references, while maintaining some that provide a handy connection between the statements and the results.

The paper primarily focuses on S6A and utilizes only a small batch of COSMIC-2 data. Section 4.2 points out a larger standard deviation in the bending angle statistics for COSMIC-2 compared to S6A, attributed to a POD solution of lower quality. It is argued that this is caused by the absence of a satellite macro-model, the lower orbit altitude, and increased solar activity in 2023. While Section 3.1 extensively discusses S6A POD, there is a lack of discussion on COSMIC-2 POD in this dedicated POD section. It is advised that the authors add a short discussion of COSMIC-2 POD, including relevant numbers or references, in order to provide a complete analysis and to support their assessment of lower COSMIC-2 POD quality in Section 4.2.

We did not expand on the C2 POD for two reasons. First, the focus here is S6A, which provide the bulk of the data. C2 is used to verify that the effect of the different GNSS clock rate is the same. With this, one can safely argue that using a higher rate clock product is beneficial, even though in absolute numbers the standard deviation for the two missions is different. Second, while for S6A there are several POD solutions that can be used as comparison (which we included in Figure 4, thanks to your previous round of comments), for C2 we are only aware of the NRT solutions available on the UCAR server. Here's a couple of examples comparing our POD solutions to UCAR's NRT orbits for the satellite with SP3 ID L77 (C2E2), for 2022 and 2024 (i.e., before and after the 2023 period analysed in the paper):